# Oral resveratrol in adults with knee osteoarthritis: A randomized placebo-controlled trial (ARTHROL)

Christelle Nguyen[1,2,3]*, Emmanuel Coudeyre[4], Isabelle Boutron[1,5,6], Gabriel Baron[5], Camille Daste[1,2,6], Marie-Martine Lefèvre-Colau[1,2,7,8], Jérémie Sellam[9,10], Jennifer Zauderer[2], Francis Berenbaum[9,10], François Rannou[1,2,3]

**1** Université Paris Cité, Faculté de Santé, UFR de Médecine, Paris, France, **2** AP-HP. Centre-Université Paris Cité, Service de Rééducation et de Réadaptation de l'Appareil Locomoteur et des Pathologies du Rachis, Hôpital Cochin, Paris, France, **3** INSERM UMR-S 1124, Toxicité Environnementale, Cibles Thérapeutiques, Signalisation Cellulaire et Biomarqueurs (T3S), Campus Saint-Germain-des-Prés, Paris, France, **4** Centre Hospitalo-Universitaire de Clermont-Ferrand, Service de Médecine Physique et de Réadaptation, INRAE, Université Clermont Auvergne, Clermont-Ferrand, France, **5** Centre d'Epidémiologie Clinique, AP-HP, Hôpital Hôtel Dieu, Paris, France, **6** INSERM UMR-S 1153, METHODS Team, Centre de Recherche Épidémiologie et Statistique Sorbonne Paris Cité, Paris, France, **7** INSERM UMR-S 1153, ECaMO Team, Centre de Recherche Épidémiologie et Statistique Sorbonne Paris Cité, Paris, France, **8** Fédération pour la Recherche sur le Handicap et l'Autonomie, Paris, France, **9** Sorbonne Université, AP-HP Hôpital Saint-Antoine, Service de Rhumatologie, Paris, France, **10** INSERM UMR_S 938, Paris, France

* christelle.nguyen2@aphp.fr

## Abstract

### Background

Resveratrol is a natural compound found in red wine. It has demonstrated anti-inflammatory properties in preclinical models. We compared the effect of oral resveratrol in a new patented formulation to oral placebo for individuals with painful knee osteoarthritis.

### Methods and findings

ARTHROL was a double-blind, randomized, placebo-controlled, Phase 3 trial conducted in 3 tertiary care centers in France. We recruited adults who fulfilled the 1986 American College of Rheumatology criteria for knee osteoarthritis and reported a pain intensity score of at least 40 on an 11-point numeric rating scale (NRS) in 10-point increments (0, no pain, to 100, maximal pain). Participants were randomly assigned (1:1) by using a computer-generated randomization list with permuted blocks of variable size (2, 4, or 6) to receive oral resveratrol (40 mg [2 caplets] twice a day for 1 week, then 20 mg [1 caplet] twice a day; resveratrol group) or matched oral placebo (placebo group) for 6 months. The primary outcome was the mean change from baseline in knee pain on a self-administered 11-point pain NRS at 3 months. The trial was registered at ClinicalTrials.gov: (NCT02905799).

Between October 20, 2017 and November 8, 2021, we assessed 649 individuals for eligibility, and from November 9, 2017, we recruited 142 (22%) participants (mean age 61.4 years [standard deviation (SD) 9.6] and 101 [71%] women); 71 (50%) were randomly assigned to the resveratrol group and 71 (50%) to the placebo group. At baseline, the mean

**Data Availability Statement:** The full original protocol, dataset and statistical codes can be accessed by academic researchers by contacting

Dr. Laëtitia Peaudecerf (laetitia.peaudecerf@aphp.fr). Data collected for the study, including individual participant data and a data dictionary defining each field in the set, will be made available to others. Deidentified participant data and a data dictionary will be made available. The study protocol and statistical analysis plan are available in the S1 and S2 Methods. Data will be shared without investigator support, after approval of a proposal, with a signed data access agreement, for research purposes.

**Funding:** This work was supported by the Ministère des Solidarités et de la Santé (Programme Hospitalier de Recherche Clinique National 2015: project no. 15-15-0234 to FR). The funders had no role in study design, data collection and analysis, decision to publish, or preparation of the manuscript.

**Competing interests:** CN is an Academic Editor on PLOS Medicine's editorial board. She reports receiving consulting fees from Thuasne, Merz and Ipsen; speaker fees from Actelion Pharmaceuticals France, Ipsen, Lilly, Meda pharma and Novartis; reimbursement of conference registration and accommodation by Grünenthal and Merz; and hospitality from Preciphar, Sandoz, Takeda France, and UCB Pharma SA, outside of the submitted work (<$5,000/y). CD reports receiving hospitality from Merz, outside of the submitted work. JS reports receiving personal fees from MSD, Pfizer, Abbvie, Fresenius Kabi, BMS, Roche, Chugai, Sandoz, Lilly, Novartis, Galapagos, AstraZeneca, UCB and Janssen and research grants from Pfizer, MSD, Schwa Medico, and BMS, outside of the submitted work. FB reports receiving consulting or speaker fees from AstraZeneca, Boehringer Ingelheim, Cellprothera, Galapagos, Grünenthal, GSK, Eli Lilly, MerckSerono, Nordic Bioscience, Novartis, Pfizer, Sanofi, Servier, Peptinov, Viatris, Aché Lab. Shareholder of 4Moving Biotech and 4P Pharma, outside of the submitted work. EC, IB, GB, MMLC, JZ and FR report no conflict of interest.

**Abbreviations:** ANCOVA, analysis of covariance; CI, confidence interval; cLDA, constrained longitudinal data analysis; NRS, numeric rating scale; OARSI, Osteoarthritis Research Society International; OMERACT, Outcome Measures in Rheumatology; SD, standard deviation; WOMAC, Western Ontario and McMaster Universities Arthritis Index.

knee pain score was 56.2/100 (SD 13.5). At 3 months, the mean reduction in knee pain was −15.7 (95% confidence interval (CI), −21.1 to −10.3) in the resveratrol group and −15.2 (95% CI, −20.5 to −9.8) in the placebo group (absolute difference −0.6 [95% CI, −8.0 to 6.9]; $p = 0.88$). Serious adverse events (not related to the interventions) occurred in 3 (4%) in the resveratrol group and 2 (3%) in the placebo group. Our study has limitations in that it was underpowered and the effect size, estimated to be 0.55, was optimistically estimated.

## Conclusions

In this study, we observed that compared with placebo, oral resveratrol did not reduce knee pain in people with painful knee osteoarthritis.

## Trial registration

ClinicalTrials.gov ID: NCT02905799.

## Author summary

### Why was this study done?

- Resveratrol has demonstrated anti-inflammatory properties in preclinical models and analgesic effects in painful conditions.

- For individuals with knee osteoarthritis, evidence before the study suggested a reduction in pain and an improvement in function at 3 months after resveratrol supplementation as an add-on therapy with meloxicam as compared with placebo. The optimal formulation of oral resveratrol was not addressed.

### What did the researchers do and find?

- We conducted a double-blind, randomized, placebo-controlled, Phase 3 trial using oral resveratrol in a new patented formulation (Patent No. WO/2010/007252).

- We recruited adults with knee osteoarthritis who reported a pain intensity score of at least 40 on an 11-point numeric rating scale (NRS) in 10-point increments (0, no pain, to 100, maximal pain).

- Participants were randomly assigned (1:1) to receive oral resveratrol or matched oral placebo for 6 months.

- Oral resveratrol did not reduce knee pain at 3 months as compared with matched oral placebo in individuals with painful knee osteoarthritis.

### What do these findings mean?

- These findings do not support the use of resveratrol supplementation for reducing knee pain in adults with painful knee osteoarthritis.

- The study has limitations in that it was underpowered and the effect size, estimated to be 0.55, was optimistically estimated.

## Introduction

Osteoarthritis is the most common cause of disability in people over 40 years old [1]. Knee osteoarthritis affects middle-aged and older individuals and results in knee pain and knee-specific activity limitations [2]. For the medium and long term, non-pharmacological treatments, including education, exercise therapy, and physical activity, are recommended as first-line treatment and have been shown to reduce pain and improve function in individuals with knee osteoarthritis [3,4]. For the short term, pharmacological treatments, including oral nonsteroidal anti-inflammatory drugs and intra-articular corticosteroids, may be considered to alleviate specific symptoms such as painful flares in selected individuals but are not recommended in the long term because of unfavorable safety profiles [5,6].

Osteoarthritis-associated chronic pain is in part driven by low-grade local and systemic inflammation [7,8]. Therefore, solutions targeting low-grade inflammation with minimal adverse effects in the long term could be of interest [9]. Resveratrol is the parent compound of a family of hydroxystilbenes existing in *cis-* and *trans-* configurations in a variety of spermatophyte plants such as grapevine, peanuts, pine, or Chinese knotweed, and found in red wine [10]. No serious toxicity has been reported, and oral resveratrol is available over the counter in many countries as a food supplement in heterogeneous formulations and dosages. In the field of rheumatic diseases, growing evidence supports the anti-inflammatory, anti-catabolic, anti-apoptotic, and anti-oxidative properties of resveratrol in different articular cell types, along with immunomodulation properties for T and B lymphocytes in vitro [11–23]. Resveratrol has also shown chondroprotective effects in vivo when injected intra-articularly in animal models of osteoarthritis [24–26]. Finally, resveratrol is believed to contribute to the benefits of the Mediterranean diet and to the French paradox [27], that is, the observation of low rates of death from coronary heart disease despite high intake of dietary cholesterol and saturated fat. However, as pointed out by some authors, such a paradox could also be, at least partially, the result of collider stratification bias [28,29].

In clinical research, the data supporting the use of resveratrol for knee pain is scanty and of low validity. In 2 trials of postmenopausal women using 2 capsules a day containing 75 mg of >98% *trans*-resveratrol, Wong and colleagues and Thaung and colleagues found no significant difference in pain and no change in pain levels from baseline in the resveratrol group with an increase in the placebo group, respectively [30,31]. In a trial of individuals with knee osteoarthritis ($N = 110$) using 1 capsule a day containing 500 mg resveratrol, Hussain and colleagues reported a reduction in pain and an improvement in function at 1 month after resveratrol supplementation as an add-on medication with meloxicam, as compared to placebo [32]. The doses used in these trials were variable and not adjusted for the low bioavailability of hydroxystilbenes. In 2010, the Yvery laboratory (Marseille, France) patented a soluble galenic form to overcome the low digestive absorption of *trans*-resveratrol as a dry powder (patent no. WO/2010/007252). In a crossover study conducted in partnership with our academic group, the plasmatic peak of *trans*-resveratrol and its metabolites was 10-fold increased in 15 healthy volunteers receiving 40 mg *trans*-resveratrol in the soluble formulation (caplets) as compared

with the original powder (capsules). The blood concentration also remained at significant levels for several hours with this soluble formulation [33].

In the current study, we assessed whether resveratrol supplementation in this new patented formulation, as an add-on therapy to usual care, could reduce knee pain at 3 months as compared with matched placebo in individuals with painful knee osteoarthritis.

## Methods

### Study design

ARTHROL was a double-blind, randomized, placebo-controlled, Phase 3 trial conducted in 3 tertiary care centres in France (Cochin and Saint-Antoine Hospitals, Paris and Gabriel-Montpied Hospital, Clermont-Ferrand). Participants were recruited among inpatients and outpatients of the physical and rehabilitation medicine and rheumatology departments. We started recruitment on November 9, 2017, and follow-up was completed on May 12, 2022. ARTHROL is reported in accordance with the CONSORT statement (S1 Appendix) [34,35]. No changes in inclusion criteria or outcomes occurred after trial commencement. The protocol of the study was approved by the *Comité de Protection des Personnes Île-de-France III* on October 26, 2017 (no. Am8977-6-3447). The protocol was published before the enrolment of the first patient, in 2017 [36]. Written informed consent was obtained from all participants. The original and final versions of the protocol and statistical analysis plan are available in the S1 and S2 Methods. All amendments to the original protocol were approved by our institutional review board and are reported in S2 Appendix. ARTHROL was registered with ClinicalTrials.gov (NCT02905799) before trial commencement. All amendments to registration on ClinicalTrials.gov are reported in S3 Appendix.

### Participants

The inclusion criteria were assessed by 6 board-certified specialists in physical and rehabilitation medicine and/or rheumatology with experience as trialists in osteoarthritis. Individuals were eligible for inclusion if they were at least 40 years old, reported pain involving the knee, reported pain duration of at least 1 month and a pain intensity score of at least 40 on a self-administered 11-point pain numeric rating scale (NRS) in 10-point increments (0, no pain, to 100, maximal pain) on the day of assessment, had X-ray evidence of knee osteoarthritis with Kellgren and Lawrence grades 1, 2, or 3 on X-rays, and fulfilled the 1986 American College of Rheumatology classification criteria for knee osteoarthritis. We chose to include Kellgren and Lawrence grades 1, 2, or 3 on X-rays because: (1) we wanted to reflect the use of dietary supplements that are offered independently of structural damages; (2) our therapeutic target was pain and not structure; and (3) there is no consistent correlation between Kellgren and Lawrence grades and pain intensity. Exclusion criteria were history of inflammatory or crystal-associated rheumatic disease; neurological disorders involving the lower limbs; knee trauma or intra-articular treatments for up to 2 months; knee surgery for up to 1 year; contraindication and/or hypersensitivity to resveratrol; current use of anticoagulants or intramuscular, intravenous, and/or oral corticosteroids; uncontrolled diseases that may require intramuscular, intravenous, and/or oral corticosteroids; participation in another biomedical research; and inability to speak, read, and/or write French. Individuals excluded for temporary reasons could be rescreened.

## Randomization and masking

Participants were randomly assigned (1:1) to receive oral resveratrol (resveratrol group) or matched oral placebo (placebo group). An independent statistician (GB) from the *Centre d'Épidémiologie Clinique* (Hôpital Hôtel-Dieu, AP-HP, Paris, France) provided a computer-generated randomization list stratified by center, with permuted blocks of variable size (2, 4, or 6). Randomization involved use of a secured software (CleanWeb, Telemedicine Technologies SAS, Boulogne-Billancourt, France). Participants, investigators, statisticians, and treating physicians were masked to the allocation group. The Yvery laboratory, which supplied the caplets of oral resveratrol and matched oral placebo, was masked to the randomization and had no contact with participants, investigators, statisticians, or treating physicians. Caplets for both the resveratrol group and placebo group had identical presentations (i.e., size, color, and taste).

## Interventions

Participants allocated to the resveratrol group received 40 mg (2 caplets) of resveratrol administered orally twice a day, 30 min before a meal with a glass of water, for 1 week, then 20 mg (1 caplet) twice a day for a total of 6 months. Resveratrol was supplied by the Yvery laboratory (patent no. WO/2010/007252). This dose was selected based on our previous study on pharmacokinetics, bioavailability, and toxicity of this formulation, showing that 40 mg of the soluble resveratrol was well absorbed and elicited biologically efficient blood levels (0.1 to 6 µm) for several hours [33]. Participants allocated to the placebo group received matched oral placebo. The placebo was also supplied by the Yvery laboratory, which ensured that caplets had identical presentations in the resveratrol and placebo groups. Participants in both groups were instructed to store the caplets in their original packaging at room temperature, with protection from humidity, light, and excessive heat. Participants were asked to return the pillboxes for caplet counts at the 3- and 6-month visits. Overall, 392 caplets were necessary for the whole duration of the study, but more caplets were supplied (i.e., 420). No specific measures to enhance adherence to the interventions were implemented, but the number of remaining caplets was counted and recorded in the electronic case report form at 6 months. The treating physician was allowed to prescribe non-pharmacological and pharmacological co-interventions as needed in both groups, including nonsteroidal anti-inflammatory drugs and analgesics. These were reported by the participant using a standardized checklist to be recorded in the electronic case report form at 3 and 6 months. No guidance was given to treating physicians and participants to control the use of analgesics and/or anti-inflammatory drugs, because resveratrol was used as an add-on therapy to usual care in the present study.

## Outcomes

We selected our primary and secondary efficacy outcomes in accordance with the Outcome Measures in Rheumatology (OMERACT) [37] and Osteoarthritis Research Society International (OARSI) recommendations [38] for Phase 3 clinical trials of knee osteoarthritis. The primary efficacy outcome was the mean change from baseline in knee pain in the last 48 h on a self-administered 11-point pain NRS at 3 months after randomization. The 11-point NRS is in 10-point increments from 0 to 100, with 0 indicating "no pain" and 100 "maximum pain." We selected our 3-month primary efficacy outcome in accordance with the recommendations of the European Society on Clinical and Economic Aspects of Osteoporosis, Osteoarthritis and Musculoskeletal Diseases for fast-acting drugs [39], which take into account guidelines from the regulatory agencies (i.e., US Food and Drug Administration and European Medicines Agency) [40,41]. Secondary efficacy outcomes were the mean change from baseline in knee pain at 6 months according to the function subscore of the self-administered Western Ontario and

McMaster Universities Arthritis Index (WOMAC) (0, no limitations, to 68, maximal limitations) at 3 and 6 months [42] and in patient global assessment on a self-administered 11-point global assessment NRS in 10-point increments (0, worst possible, to 100: best possible) at 3 and 6 months; the proportion of responders according to the OARSI-OMERACT at 3 and 6 months [43]; and the number of intra-articular injections of corticosteroids or hyaluronic acid and consumption of analgesics and nonsteroidal anti-inflammatory drugs reported on a self-administered four-category scale (i.e., never, several times a month, several times a week, or daily) at 3 and 6 months and dichotomized for analysis as never versus all other responses (dichotomization was prespecified before analysis). To minimize the data collectors' influence on participants' answers, participants were instructed to complete self-administered questionnaires from home at 3 months and 6 months, before the scheduled follow-up visits. Safety outcomes were assessed by the investigator by asking an open-ended question, "Have you had any adverse events since last contact?", at 3 and 6 months. The investigator assessed the causality relation between the adverse event and the administered treatment using the World Health Organisation-Uppsala Monitoring Centre method. All outcomes prespecified in the protocol were reported.

## Statistical analysis

The original and final versions of the statistical analysis plan are available in the S2 Method. With an α risk of 0.05, power (1-β) of 0.90, and predicted mean difference in mean change in knee pain at 3 months of 15 (standard deviation (SD) 27) points, which corresponds to an effect size of 0.55, we needed 69 participants in each group. Estimating that 15% of participants would be lost to follow-up, we sought to include 82 participants in each group. Categorical variables are described with frequencies and percentages and quantitative variables with mean (SD).

To compare between-group differences in mean change for quantitative outcomes (knee pain, WOMAC function, and patient global assessment), we used a constrained longitudinal data analysis (using the REstricted Maximum Likelihood algorithm, *REML*) [44–46]. This mixed model is a constrained full-likelihood approach, whereby both the baseline and post-baseline values are modeled as dependent variables (the constrained longitudinal data analysis model assumes that both the baseline and post-baseline measurements are jointly multivariate normally distributed because the baseline value is treated as part of the response vector). The true baseline means are constrained to be the same for the 2 treatment groups. The constrained longitudinal data analysis model can include all randomized participants with at least 1 baseline or post-baseline value. Such methods based on maximum likelihood are consistent under the missing-at-random assumption. Hence, this analysis provides an adjustment for the observed baseline difference in estimating the treatment. Differences in mean change from baseline with 95% confidence intervals (CIs) at 3 and 6 months after randomization and mean change from baseline in one of the 2 groups were directly estimated by parameters of the model. Mean change from baseline in the remaining group was derived by linear combination. As a sensitivity analysis, we also analyzed primary outcome with constrained longitudinal data analysis (cLDA) model when considering only baseline and 3 months data and with classical analysis of covariance (ANCOVA). To assess the impact of missing data, we also have considered Worst-case scenario and Best-case scenario in the framework of ANCOVA model. Worst-case scenario assumes missing NRS knee pain values at 3 months in Resveratrol group had the worst possible value (= 100) and those in control group had the best possible value (= 0). Best-case scenario assumes missing NRS knee pain values at 3 months in Resveratrol group had the best possible value (= 0) and those in control group had the worst possible value (= 100). For the primary outcome, we also report a cumulative analysis graph for percentage of responders with relative change in NRS knee pain score at 3 months [47].

The self-reported number of intra-articular injections of corticosteroids or hyaluronic acid at 3 and 6 months was dichotomized in injections (yes/no) after examining distribution (89% of values equal to 0). Consequently, all remaining outcomes were dichotomous. For these outcomes, a Poisson model with log link under regression standardization framework allowed for estimating the marginal measure of association. Results are expressed as absolute differences in proportions between groups, relative risk, and 95% CIs at 3 and 6 months after randomization. Because we had only 3 centers, we did not use generalized estimating equation approach as planned, but resorted on models with center as a fixed effect for quantitative and dichotomous outcomes, as recommended [48]. Safety outcomes were described in each group. All statistical tests were 2-sided, with $P < 0.05$ considered statistically significant. Data were analyzed by using SAS 9.4 (SAS Institute) with the procedure MIXED (constrained longitudinal data analysis model). The other analyses involved using R 4.1.1 (R Foundation for Statistical Computing). The R package stdReg was used for the log Poisson model.

## Results

### Participants

Between October 20, 2017 and November 8, 2021, we assessed 649 individuals for eligibility and from November 9, 2017, we recruited 142 (22%) participants; 71 (50%) were randomly assigned to the resveratrol group and 71 (50%) to the placebo group. The study was stopped before the prespecified number of participants was reached (i.e., 164), because of slow accrual during the 2020 to 2022 period (COVID-19 pandemic) and lack of further funding. Overall, 120/142 (85%) participants completed the allocated intervention (Fig 1). The mean age of participants was 61.4 years (SD 9.6), 101 (71%) were females (Table 1). At baseline, the mean knee pain score was 56.2/100 (SD 13.5) and mean duration of symptoms was 8.5 years (8.2) (Table 1).

### Primary outcome

Baseline characteristics of patients with missing data on primary efficacy outcome data ($n = 7$) were and those with complete data ($n = 135$) were reported in S4 Appendix. The estimated differences between groups for the primary outcome were small, with wide confidence intervals (−15.7 [95% CI, −21.1 to −10.3] versus −15.2 [95% CI, −20.5 to −9.8]; absolute difference −0.6 [95% CI, −8.0 to 6.9]; $p = 0.88$) at 3 months (Table 2). Results of sensitivity analysis were reported in S5 Appendix. The evolution of knee pain during follow-up is shown in Fig 2. The cumulative analysis graph of proportion of responders with relative change in knee pain at 3 months (Fig 3) also illustrates similar range of response levels between the groups. For instance, when considering a response level of at least 20%, corresponding to the OARSI-OMERACT response for pain [43], the responder rate was 54% in the resveratrol group and 56% in the placebo group.

### Secondary outcomes

The estimated differences for knee pain at 6 months were small, with wide CIs (absolute difference 0.4 [95% CI, −8.4 to 9.1]; $p = 0.93$) (Table 2). At 3 and 6 months, the OARSI-OMERACT response was 52% (34/66 participants) and 48% (29/60 participants), respectively, in the resveratrol group and 50% (34/68 participants) and 52% (34/66 participants) in the placebo group (Table 2). The estimated differences between groups or relative risks for rescue medication since last contact, including intra-articular corticoids and/or hyaluronan, non-opioid analgesics, opioid analgesics, and nonsteroidal anti-inflammatory drugs at 3 and 6 months (Table 2) or any other secondary outcomes, were small, with wide CI (Table 2).

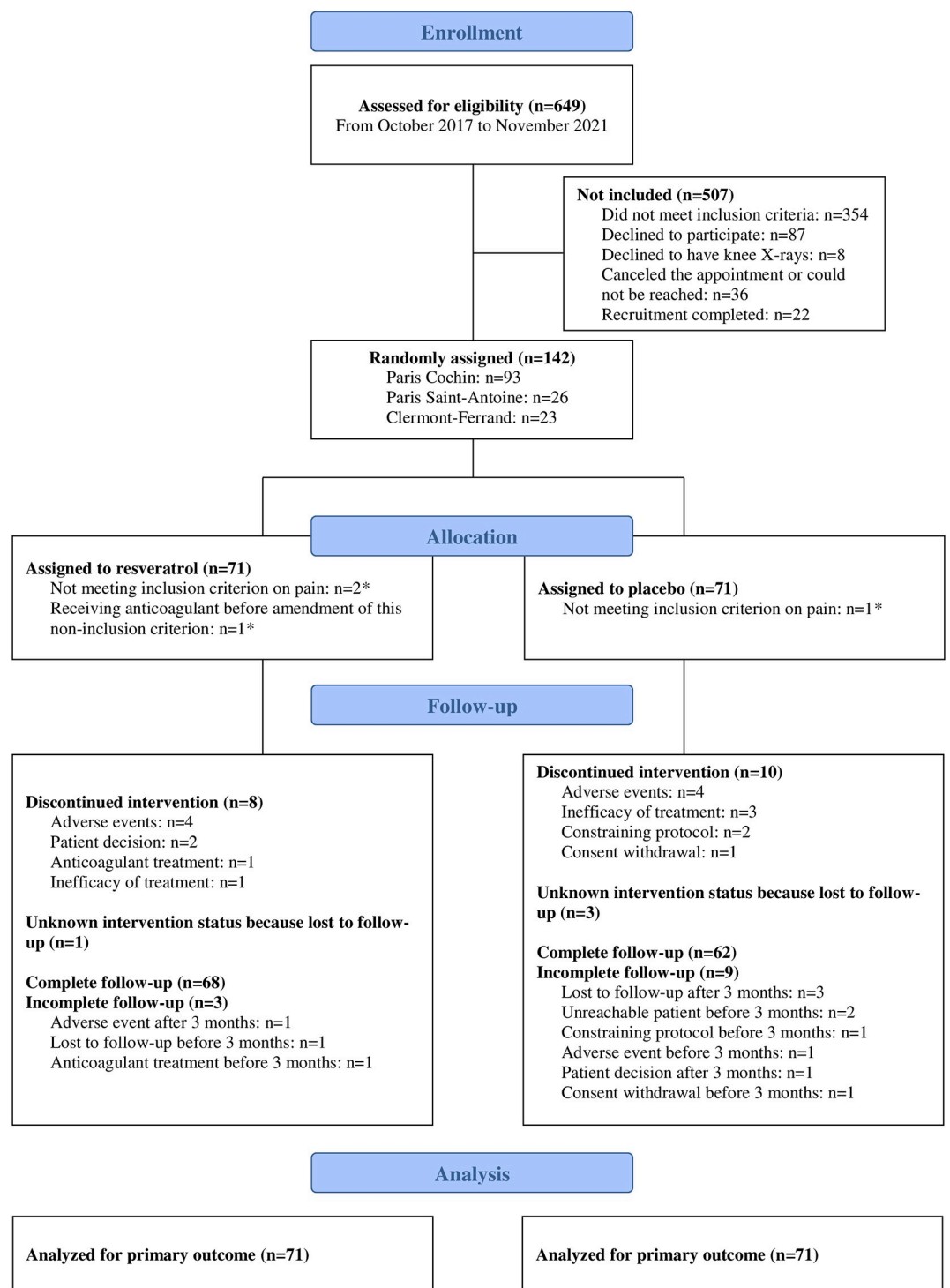

**Fig 1. Enrolment, randomization, and follow-up.**

**Table 1. Demographic and clinical characteristics of participants.**

| | Resveratrol n = 71 | Placebo n = 71 | Total n = 142 |
|---|---|---|---|
| Age (years), mean (SD) | 59.8 (8.9) | 63.0 (10.1) | 61.4 (9.6) |
| Women, n (%) | 50 (70) | 51 (72) | 101 (71) |
| Body mass index (kg/m$^2$), mean (SD) | 28.3 (6.7; n = 70) | 28.3 (5.6) | 28.3 (6.2; n = 141) |
| Higher education, n (%) | 47/71 (66) | 48/71 (68) | 95/142 (67) |
| Employment status, n (%) | | | |
| • Full- or part-time employment | 39 (55) | 33 (47) | 72 (51) |
| • Sick leave | 3 (4) | 0 (0) | 3 (2) |
| • Unable to work | 1 (1) | 3 (4) | 4 (3) |
| • Retired | 28 (39) | 35 (49) | 63 (44) |
| Treatments in the previous 3 months, n (%) | | | |
| • Intra-articular corticoids and/or hyaluronan | 13/70 (19) | 7/70 (10) | 20/140 (14) |
| • Non-opioid oral analgesics | 40/68 (59) | 46/69 (67) | 86/137 (63) |
| • Weak opioid oral analgesics* | 14/64 (22) | 20/67 (30) | 34/131 (26) |
| • Strong opioid oral analgesic* | 1/61 (2) | 1/63 (2) | 2/124 (2) |
| • Oral nonsteroidal anti-inflammatory drugs | 32/70 (46) | 29/70 (31) | 61/140 (44) |
| • Symptomatic slow-acting drugs for osteoarthritis | 11/70 (16) | 8/70 (11) | 19/140 (14) |
| • Physiotherapy | 20 (28) | 26 (37) | 46 (32) |
| • Home-based exercises | 27 (38) | 31 (44) | 58 (41) |
| • Foot insoles | 32 (45) | 32 (45) | 64 (45) |
| • Knee brace | 17 (24) | 15 (21) | 32 (23) |
| • Walking aids | 9 (13) | 4 (6) | 13 (9) |
| • Weight management | 30 (42) | 22 (31) | 52 (37) |
| Clinical characteristics, mean (SD) | | | |
| • Knee pain intensity (NRS, 0–100)§ | 56.9 (14.0) | 55.5 (13.1) | 56.2 (13.5) |
| • Knee pain duration (years) | 8.2 (7.6; n = 70) | 8.9 (8.7) | 8.5 (8.2; n = 141) |
| • WOMAC function subscore (0–68)‖ | 44.1 (16.0) | 44.4 (16.9) | 44.2 (16.4) |
| • Patient global assessment\(NRS, 0–100)¶ | 69.2 (20.1) | 63.0 (22.0) | 66.1 (21.2) |
| X-ray findings in medial or lateral femorotibial or patellofemoral, n (%) | | | |
| • Maximal KL grade 1 | 13 (18) | 11 (16) | 24 (17) |
| • Maximal KL grade 2 | 22 (31) | 23 (32) | 45 (32) |
| • Maximal KL grade 3 | 36 (51) | 37 (52) | 73 (51) |

*Weak opioids include codeine, dihydrocodeine, and tramadol. Strong opioids include morphine, diamorphine, fentanyl, buprenorphine, oxymorphone, oxycodone, and hydromorphone.

§Higher scores indicate greater pain.

‖Higher scores indicate more limitations.

¶Higher scores indicate better health.

n = 71 per group unless indicated otherwise.

KL, Kellgren and Lawrence; NRS, numeric rating scale; SD, standard deviation; WOMAC, Western Ontario and McMaster Universities Osteoarthritis Index.

## Safety

During follow-up, a total of 95 adverse events were reported in both groups: minor adverse events or serious adverse events were reported in 41% (29/71 participants) and 42% (30/71) in resveratrol (n = 52 events) and placebo (n = 43 events) groups, respectively (Table 3). Overall, 7 serious adverse events were reported in the 2 groups: 4 events for 3 participants (4%) in the

**Table 2. Primary and secondary efficacy outcomes.**

| Outcome | Resveratrol<br>n = 71 | Placebo<br>n = 71 | Absolute difference (resveratrol minus placebo)<br>(95% CI) | Relative risk (resveratrol vs. placebo)<br>(95% CI) | p-Value |
|---|---|---|---|---|---|
| **Primary efficacy outcome** | | | | | |
| **3 months after randomization** | | | | | |
| • Change in knee pain (NRS, 0–100), mean (95% CI)[§] | −15.7 (−21.1 to −10.3) | −15.2 (−20.5 to −9.8) | −0.6 (−8.0 to 6.9) | - | 0.88 |
| **Secondary efficacy outcomes** | | | | | |
| **3 months after randomization** | | | | | |
| • Change in WOMAC function subscore (0–68), mean (95% CI)[‖] | −9.2 (−13.0 to −5.4) | −10.6 (−14.3 to −6.8) | 1.4 (−3.9 to 6.7) | - | 0.59 |
| • Change in PGA (NRS, 0–100), mean (95% CI)[¶] | 1.4 (−3.3 to 6.2) | 1.2 (−3.5 to 5.9) | 0.2 (−5.9 to 6.4) | - | 0.95 |
| • OARSI-OMERACT response, n (%) | 34/66 (52) | 34/68 (50) | 1.5 (−15.3 to 18.3) | 1.03 (0.74 to 1.43) | 0.86 |
| • Intra-articular corticoids and/or hyaluronan since last contact | 5/67 (8) | 6/67 (9) | −1.6 (−10.7 to 7.5) | 0.82 (0.27 to 2.51) | 0.73 |
| • Non-opioid analgesics since last contact, n (%) | 38/67 (57) | 39/64 (61) | −4.5 (−21.4 to 12.4) | 0.93 (0.70 to 1.24) | 0.60 |
| • Weak opioid analgesics since last contact, n (%)* | 12/62 (19) | 14/64 (22) | −2.4 (−16.5 to 11.7) | 0.89 (0.45 to 1.76) | 0.74 |
| • Strong opioid analgesics since last contact, n (%)* | 1/62 (2) | 1/60 (2) | −0.2 (−4.9 to 4.5) | 0.90 (0.05 to 15.55) | 0.94 |
| • Nonsteroidal anti-inflammatory drugs since last contact, n (%) | 18/66 (27) | 24/67 (36) | −8.9 (−24.4 to 6.8) | 0.75 (0.46 to 1.25) | 0.27 |
| **6 months after randomization** | | | | | |
| • Change in knee pain (NRS, 0–100), mean (95% CI)[§] | −16.8 (−23.4 to −10.3) | −17.1 (−23.4 to −10.9) | 0.4 (−8.4 to 9.1) | - | 0.93 |
| • Change in WOMAC function subscore (0–68), mean (95% CI)[‖] | −12.6 (−17.3 to −8.0) | −9.4 (−14.0 to −4.9) | −3.2 (−9.5 to 3.1) | - | 0.32 |
| • Change in PGA (NRS, 0–100), mean (95% CI)[¶] | 1.8 (−4.2 to 7.9) | 1.9 (−3.9 to 7.8) | −0.2 (−7.7 to 7.5) | - | 0.98 |
| • OARSI-OMERACT response, n (%) | 29/60 (48) | 34/66 (52) | −3.6 (−21.1 to 13.9) | 0.93 (0.74 to 1.43) | 0.68 |
| • Intra-articular corticoids and/or hyaluronan since last contact | 7/60 (12) | 5/65 (8) | 4.0 (−6.2 to 14.1) | 1.51 (0.55 to 4.39) | 0.44 |
| • Non-opioid analgesics since last contact, n (%) | 30/59 (51) | 33/63 (52) | −2.6 (−20.2 to 15.0) | 0.95 (0.68 to 1.34) | 0.77 |
| • Weak opioid analgesics since last contact, n (%)* | 9/60 (15) | 17/62 (27) | −12.3 (−26.5 to 1.9) | 0.55 (0.27 to 1.13) | 0.09 |
| • Strong opioid analgesics since last contact, n (%)* | 1/60 (2) | 1/60 (1) | 0.0 (−4.6 to 4.5) | 0.97 (0.06 to 15.25) | 0.99 |
| • Nonsteroidal anti-inflammatory drugs since last contact, n (%) | 15/60 (25) | 20/65 (31) | −6.5 (−22.0 to 9.0) | 0.79 (0.45 to 1.39) | 0.41 |

*Weak opioids include codeine, dihydrocodeine, and tramadol. Strong opioids include morphine, diamorphine, fentanyl, buprenorphine, oxymorphone, oxycodone, and hydromorphone.

[§]Higher scores indicate greater pain, n = 71 in resveratrol group and n = 71 in placebo group at baseline, n = 67 and n = 68 at 3 months, n = 60 and n = 66 at 6 months.

[‖]Higher scores indicate more limitations, n = 71 in resveratrol group and n = 71 in placebo group at baseline, n = 66 and n = 67 at 3 months, n = 60 and n = 65 at 6 months.

[¶]Higher scores indicate better health, n = 71 in resveratrol group and n = 71 in placebo group at baseline, n = 67 and n = 68 at 3 months, n = 60 and n = 66 at 6 months.

CI, confidence interval; PGA, patient global assessment; NRS, numeric rating scale; SD, standard deviation; WOMAC, Western Ontario and McMaster Universities Osteoarthritis Index; OARSI-OMERACT, Outcome Measures in Rheumatology-Osteoarthritis Research Society International.

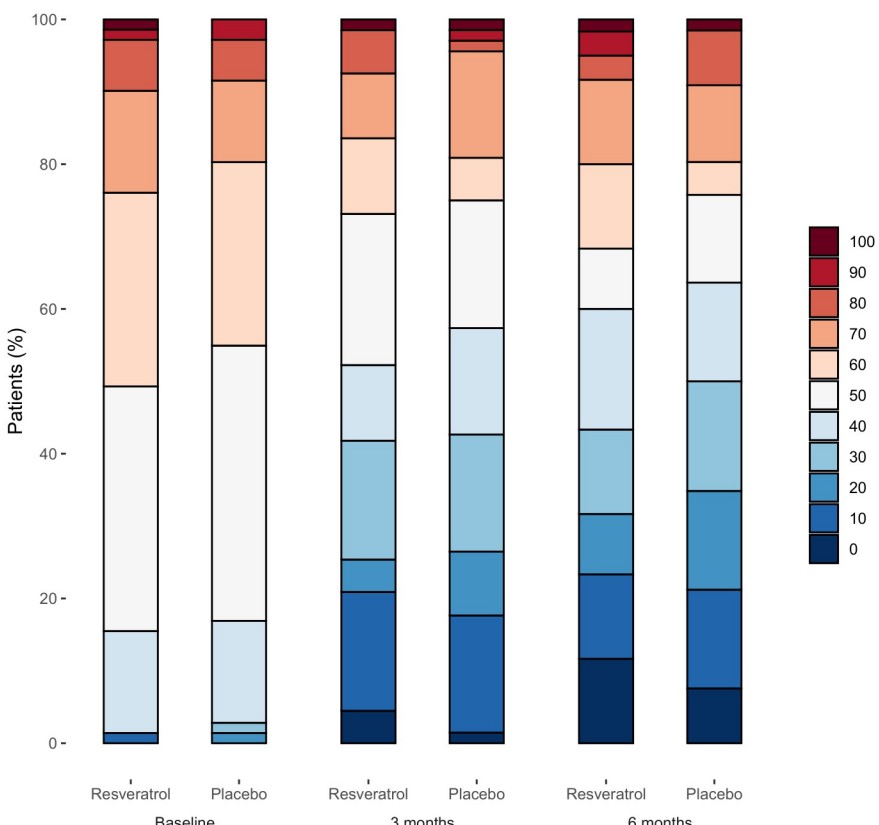

**Fig 2. Distribution of numeric rating scale knee pain intensity (0–100) at baseline, 3 and 6 months (*n* = 71 in resveratrol group and *n* = 71 in placebo group at baseline, *n* = 67 and *n* = 68 at 3 months, *n* = 60 and *n* = 66 at 6 months).** The scale is an 11-point numeric rating scale in 10-point increments (0, no pain, to 100, maximal pain).

resveratrol group and 3 events for 2 participants (3%) in the placebo group. No events were considered related to the interventions.

## Adherence to interventions

At the end of the study, the mean number of remaining caplets was 239.4 (SD 49.1, *n* = 58) in the resveratrol group and 222.2 (SD 27.6, *n* = 48) in the placebo group. We observed no imbalance in non-pharmacological and pharmacological co-interventions, as usual care, between the 2 groups at 3 and 6 months (S5 Appendix).

## Discussion

In this randomized placebo-controlled trial of oral resveratrol for painful knee osteoarthritis, we found no evidence of a greater reduction in knee pain in the resveratrol than placebo group at 3 and 6 months. Therefore, one can assume that oral resveratrol may not be effective in this indication and may not have a biologic effect on the pain pathway, but some other reasons may explain our results.

First, even though preclinical data suggested the anti-inflammatory and chondroprotective properties of resveratrol in vitro and in vivo [9] and an optimized formulation of resveratrol was used for the present study [33], oral resveratrol at the doses and in the formulation we used failed to improve relevant clinical outcomes for individuals with painful knee

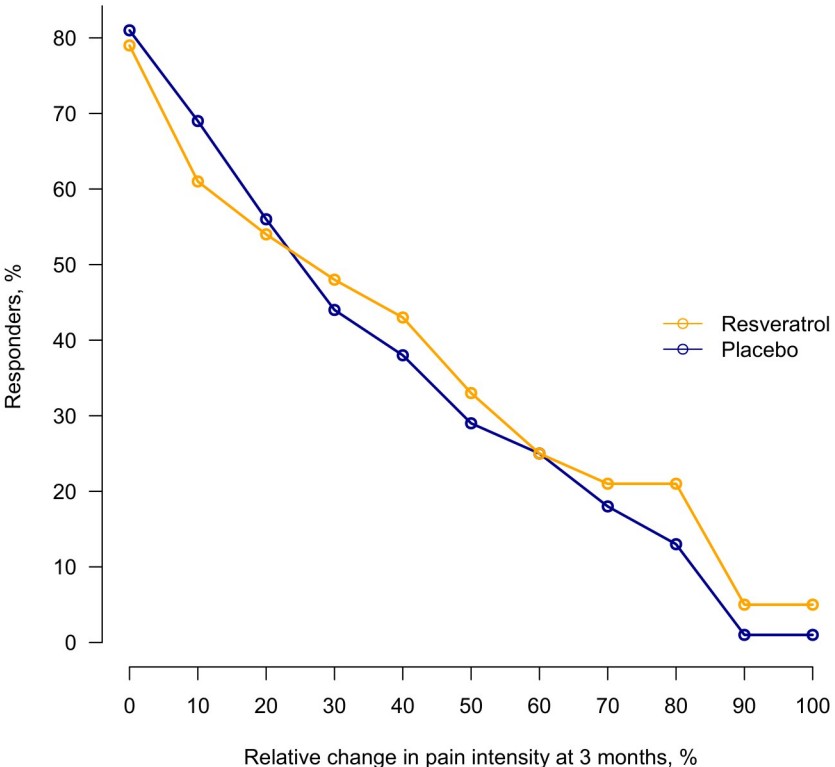

**Fig 3. Cumulative percentage of responders with relative change in numeric rating scale score for knee pain at 3 months.** At 3 months, data were available for 67 participants in the resveratrol group and 68 in the placebo group.

osteoarthritis. A reason may be insufficient bioavailability of *trans*-resveratrol reached in the targeted tissues. However, we did not collect blood or synovial fluid samples to support this hypothesis. In a previous study in humans, Marouf and colleagues reported a decrease in serum levels of biomarkers of inflammation, including interleukin-1β and interleukin-6, tumor necrosis factor, C-reactive protein, and complement proteins C3 and C4, with oral resveratrol [49], but found a nonsignificant correlation with clinical outcomes [50]. Furthermore, in these studies, variations in clinical outcomes were of the same magnitude for pain and activity limitations as we observed in ARTHROL, which suggests limited clinical effect of oral resveratrol on core outcomes of knee osteoarthritis, despite a reduction in biological biomarker levels. Overall, the question over whether the dose formulation might be considered large enough to produce a therapeutic effect remains unanswered. The pharmacokinetics study of this resveratrol formulation suggested the dose used did remain in the bloodstream of patients; however, there were no assessments of efficacy. Indeed, even though, the dose formulation was sufficient to increase anti-inflammatory markers production in the model of high-fat diet-fed C57Bl/6J wild-type mice in 3 different organs at the level of mRNA expression (i.e., liver interleukin-10, colon heme oxygenase-1, and hypothalamus plasminogen activator inhibitor-1) [33], whether these results may be translated in human inflammatory conditions has not been assessed yet. Further, some other studies used doses over 10 times that used in this trial [32].

Second, our population may be considered as having severe symptoms. At baseline, participants had long-lasting and high levels of pain and activity limitations as well as fairly severe structural damage (>40% participants with Kellgren and Lawrence grade 3 on X-rays). A

**Table 3.  Safety outcomes.**

| | Resveratrol n = 71 | Placebo n = 71 | Total n = 142 |
|---|---|---|---|
| **Patients with at least 1 serious adverse event or 1 minor adverse event** | 29/71 (41) | 30/71 (42) | 59/142 (42) |
| **Patients with at least 1 serious adverse event** | 3/71 (4) | 2/71 (3) | 5/142 (4) |
| **Total number of serious adverse events** | 4 | 3 | 7 |
| • Hospitalization for another reason | 4 | 3 | 7 |
| **Patients with at least 1 minor adverse event** | 28/71 (39) | 28/71 (39) | 56/142 (39) |
| **Total number of minor adverse events** | 48 | 40 | 88 |
| • Other musculoskeletal pain | 10 | 8 | 18 |
| • Knee pain | 10 | 3 | 13 |
| • Hospitalization for another reason | 5 | 6 | 11 |
| • Abdominal pain | 4 | 5 | 9 |
| • Diarrhea/nausea | 6 | 2 | 8 |
| • Rhinitis | 2 | 5 | 7 |
| • SARS-CoV-2 infection | 0 | 3 | 3 |
| • Headache | 2 | 0 | 2 |
| • Fatigue | 1 | 1 | 2 |
| • Sinusitis | 1 | 1 | 2 |
| • Anemia | 1 | 0 | 1 |
| • Fall | 0 | 1 | 1 |
| • Dyslipidemia | 0 | 1 | 1 |
| • Dysphagia | 1 | 0 | 1 |
| • High blood pressure | 1 | 0 | 1 |
| • Insomnia | 1 | 0 | 1 |
| • Drug interactions | 0 | 1 | 1 |
| • Leukemia | 0 | 1 | 1 |
| • Paresthesia | 0 | 1 | 1 |
| • Bronchitis | 1 | 0 | 1 |
| • Vertigo | 1 | 0 | 1 |

Numbers are absolute frequencies or n (%).

more comprehensive and multidisciplinary therapeutic approach including a bundle of pharmacological and non-pharmacological treatments may be more appropriate in this population [3,4]. However, at baseline, most of our participants reported having received non-opioid analgesics (72%), nonsteroidal anti-inflammatory drugs (44%), intra-articular injections (14%), as well as home-based exercise therapy (41%), weight management (37%), and physiotherapy (32%) in the previous 3 months. Furthermore, co-interventions were well balanced between the 2 groups. Therefore, in most participants, usual care was fairly optimal and resveratrol was offered as an add-on therapy.

Third, we observed a reduction in knee pain in both groups of approximately 17/100 points and 15/100 points at 3 and 6 months, respectively. In addition, the percentage of OARSI-O-MERACT responders in the placebo group was 50% and 52%, at 3 and 6 months, consistent with previous reports [51]. These findings may be explained by the placebo and Hawthorne effects and a regression toward the mean. They reinforce the challenge in showing that a specific treatment is superior to placebo in knee osteoarthritis and in designing properly controlled trials.

Finally, concomitant analgesia was unlimited and prescribed as necessary by the participating physicians. Therefore, an alternative explanation for the negligible between-groups effect observed is that some participants may have masked the effect of resveratrol with their concomitant analgesics. Indeed, over half the sample was taking at least 1 analgesic at 6 months.

Our study has limitations. No guidance was given to control the use of other analgesics and we did not collect data on new initiations of therapy. Therefore, we cannot exclude that both recorded and non-recorded co-interventions might have affected efficacy outcomes. Demonstration of efficacy in adjunctive treatment trials is difficult even when the on-going treatment is maintained stable and has rarely been successful when used in knee osteoarthritis studies. For an initial demonstration of efficacy, particularly with small sample sizes, variability may have needed to be kept to a minimum. In the present study, participants and their providers could change treatments as needed. Thus, a metric to demonstrate efficacy would appear to be a reduction in on-going treatment. This was difficult at best and impossible without a more granular approach to evaluating daily medication use (concomitant and study) which was not done in our study. However, allowing co-interventions in both groups aimed at reflecting the use of resveratrol as an add-on therapy to usual care. Furthermore, in the comparative analyses, the estimated relative risk for rescue medication at 3 and 6 months was small, with wide confidence intervals. Our study was significantly under-powered. The effect size being estimated to be 0.55 was optimistic since there is no oral medical intervention for knee osteoarthritis that has a standardized effect size greater than 0.4 and many are closer to 0.3. The consequences of selecting such a large effect size were a reduction in the sample size estimate which, with dropouts, was 82/group. Even this number was not achieved (142 randomized versus 164 goal) due to limitations of enrollment period secondary to COVID-19 and funding considerations. Therefore, the failure to find differences between groups in the present study did not necessarily mean that no differences would be seen if the study were adequately powered. Moreover, the number of patients screened versus those randomized was low, despite the inclusion/exclusion criteria being relatively relaxed. Finally, dropouts and/or missing data could impact the results. However, given the low missing data rate of the primary outcome (7/142, 5%) and the fact that missing data are balanced in proportion across groups (4/71 and 3/71 in resveratrol group and in placebo group, respectively), it seems likely that missing data would have a minimal impact on the estimate of the primary treatment effect. Moreover, our worst-case and best-case sensitivity analyses indicate that our conclusions are robust to a wide range of assumptions regarding this missing data.

In summary, the estimated differences between groups in mean change from baseline in knee pain at 3 and 6 months were small, with wide confidence intervals. These findings do not support the use of resveratrol supplementation for reducing knee pain in adults with painful knee osteoarthritis.

## Supporting information

**S1 Appendix. Consolidated standards of reporting trials (CONSORT) checklist.**
(DOCX)

**S2 Appendix. Amendments to the original protocol.**
(DOCX)

**S3 Appendix. Amendments to registration on ClinicalTrials.gov.**
(DOCX)

**S4 Appendix. Demographic and clinical characteristics of participants with missing data on primary efficacy outcome (PEO).**
(DOCX)

**S5 Appendix. Sensitivity analysis on primary outcome.**
(DOCX)

**S6 Appendix. Non-pharmacological and pharmacological co-interventions since last contact at 3 and 6 months.**
(DOCX)

**S1 Method. Original (version 1.1 of 14/12/2016) and final (version 7.0 of 23/03/2021) versions of the full protocol and publication of the protocol [36].**
(PDF)

**S2 Method. Original (version 1, October 3, 2022) and final (version 2, October 4, 2022) versions of the statistical analysis plan.**
(PDF)

## Acknowledgments

The authors thank Recherche Clinique, Entrepôts de Données et Pharmacologie GHU Paris Centre Université Paris Cité—Unité de Recherche Clinique (Mrs. Alexandra Bruneau, Mrs. Inès Chermak, Dr. Claire Du Ranquet, Dr. Laëtitia Peaudecerf, and Mr. Mady Diallo Traoré) for implementation, monitoring, and data management of the study; and Mrs. Laura Smales for professional copyediting. Oral resveratrol and matched oral placebo were supplied by the Yvery laboratory (Mr. Laurent Pechère and Dr. Éric Serrée) free of charge.

## Author Contributions

**Conceptualization:** Christelle Nguyen, Emmanuel Coudeyre, Isabelle Boutron, Gabriel Baron, Jérémie Sellam, Francis Berenbaum, François Rannou.

**Data curation:** Christelle Nguyen, Isabelle Boutron, François Rannou.

**Formal analysis:** Christelle Nguyen, Isabelle Boutron, Gabriel Baron, Francis Berenbaum, François Rannou.

**Funding acquisition:** Christelle Nguyen, François Rannou.

**Investigation:** Christelle Nguyen, Emmanuel Coudeyre, Isabelle Boutron, Camille Daste, Marie-Martine Lefèvre-Colau, Jérémie Sellam, Jennifer Zauderer, Francis Berenbaum.

**Methodology:** Christelle Nguyen, Isabelle Boutron, Gabriel Baron, François Rannou.

**Project administration:** Christelle Nguyen, François Rannou.

**Software:** Gabriel Baron.

**Supervision:** Christelle Nguyen, Isabelle Boutron, Jérémie Sellam, Francis Berenbaum, François Rannou.

**Validation:** Christelle Nguyen, Emmanuel Coudeyre, Isabelle Boutron, Francis Berenbaum, François Rannou.

**Visualization:** Christelle Nguyen.

**Writing – original draft:** Christelle Nguyen, Isabelle Boutron, Gabriel Baron, Jérémie Sellam, Francis Berenbaum, François Rannou.

**Writing – review & editing:** Christelle Nguyen, Emmanuel Coudeyre, Isabelle Boutron, Gabriel Baron, Camille Daste, Marie-Martine Lefèvre-Colau, Jérémie Sellam, Jennifer Zauderer, Francis Berenbaum, François Rannou.

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
