## [Editor Report · Decision Letter 0]

18 Mar 2024

Dear Dr Nguyen, 

Thank you for submitting your manuscript entitled "Oral resveratrol in adults with knee osteoarthritis: a randomized placebo-controlled trial (ARTHROL)" for consideration by PLOS Medicine.

Your manuscript has now been evaluated by the PLOS Medicine editorial staff and I am writing to let you know that we would like to send your submission out for external peer review.

Due to the clinical trial nature of your submission, we ask that you provide a copy of your trial protocol and a completed CONSORT checklist as supporting information. By protocol, we mean the complete and detailed plan for the conduct and analysis of the trial that the ethics committee approved before the trial began. Please send this in the original language. If this is in a language other than English, please also provide a translation. Please detail any deviations from this study protocol in the Methods section of your manuscript. The documents will be made available to the editors and reviewers.

Please re-submit your manuscript within two working days, i.e. by Mar 20 2024.

Feel free to email me at aschaefer@plos.org or us at plosmedicine@plos.org if you have any queries relating to your submission.

Kind regards,

Alexandra Schaefer, PhD

Associate Editor

PLOS Medicine

---

## [Decision Letter · Decision Letter 1]

25 Apr 2024

Dear Dr. Nguyen,

Thank you very much for submitting your manuscript "Oral resveratrol in adults with knee osteoarthritis: a randomized placebo-controlled trial (ARTHROL)" (PMEDICINE-D-24-00858R1) for consideration at PLOS Medicine. 

Your paper was evaluated by an associate editor and discussed among all the editors here. It was sent to independent reviewers, including a statistical reviewer. The reviews are appended at the bottom of this email and any accompanying reviewer attachments can be seen via the link below:

[LINK]

In light of these reviews, I am afraid that we will not be able to accept the manuscript for publication in the journal in its current form, but we would like to consider a revised version that addresses the reviewers' and editors' comments. Obviously we cannot make any decision about publication until we have seen the revised manuscript and your response, and we plan to seek re-review by one or more of the reviewers. 

Please use the following link to submit the revised manuscript: https://www.editorialmanager.com/pmedicine/

We expect to receive your revised manuscript by May 16 2024. However, if this deadline is not feasible, please contact me by email, and we can discuss a suitable alternative.

Don't hesitate to contact me directly with any questions (aschaefer@plos.org). If you reply directly to this message, please be sure to 'Reply All' so your message comes directly to my inbox.

We look forward to receiving your revised manuscript. 

Sincerely,

Alexandra Schaefer, PhD

PLOS Medicine

plosmedicine.org

***Please note: not all will apply to your paper, but please check each item carefully

GENERAL COMMENTS

1) Please cite the reference numbers in square brackets. Citations should be preceding punctuation.

COMPETING INTEREST

All authors must declare their relevant competing interests per the PLOS policy, which can be seen here: https://journals.plos.org/plosmedicine/s/competing-interests

For authors with ties to industry, please indicate whether any of the interests has a financial stake in the results of the current study.

DATA AVAILABILITY STATEMENT 

The Data Availability Statement (DAS) requires revision. For each data source used in your study: 

ABSTRACT

1) Please report your abstract according to CONSORT for abstracts, following the PLOS Medicine abstract structure (Background, Methods and Findings, Conclusions). https://www.equator-network.org/reporting-guidelines/consort-abstracts/]

2) PLOS Medicine requests that main results are quantified with 95% CIs as well as p values. When reporting p values please report as p<0.001 and where higher as the exact p value p=0.002, for example. For the purposes of transparent data reporting, if not including the aforementioned please clearly state the reasons why not. When a p value is given, please specify the statistical test used to determine it.

3) Throughout, suggest reporting statistical information as follows to improve clarity for the reader “22% (95% CI [13%,28%]; p</=)”. Please be sure to define all numerical values at first use. Please amend throughout the abstract and main manuscript. Please note the use of commas to separate upper and lower bounds, as opposed to hyphens as these can be confused with reporting of negative values.

4) Please ensure that all numbers presented in the abstract are present and identical to numbers presented in the main manuscript text.

5) Please include the study design, population and setting, number of participants, years during which the study took place (enrollment and follow up), length of follow up, and main outcome measures.

6) Please specify who was blinded to the intervention and control, define the intervention and control states, provide the number in each group, state that analysis was intention to treat and provide the number of participants lost to follow up in each group.

7) Please include the actual amounts and/or absolute risk(s) of relevant outcomes (including NNT or NNH where appropriate), not just relative risks or correlation coefficients (example for absolute risks: PMID: 28399126).

8) Please include the important dependent variables that are adjusted for in the analyses.

9) Please define all abbreviations including those for statistical reporting at first use.

10) In the last sentence of the Abstract Methods and Findings section, please describe the main limitation(s) of the study's methodology.

11) Please include the clinical trial registry number in the abstract.

AUTHOR SUMMARY

At this stage, we ask that you include a short, non-technical Author Summary of your research to make findings accessible to a wide audience that includes both scientists and non-scientists (Please remove the "Research in context" section). The Author Summary should immediately follow the Abstract in your revised manuscript. This text is subject to editorial change and should be distinct from the scientific abstract. Ideally each sub-heading should contain 2-3 single sentence, concise bullet points containing the most salient points from your study. In the final bullet point of ‘What Do These Findings Mean?’, please include the main limitations of the study in non-technical language. Please see our author guidelines for more information: https://journals.plos.org/plosmedicine/s/revising-your-manuscript#loc-author-summary

METHODS AND RESULTS

1) PLOS Medicine requests that main results are quantified with 95% CIs as well as p values. We suggest reporting statistical information as detailed above – see under ABSTRACT

2) Please present numerators and denominators for percentages (at least in the Tables [not necessarily each time they're mentioned]).

3) Please complete the CONSORT checklist and ensure that all components of CONSORT are present in the manuscript, including how randomization was performed, allocation concealment, blinding of intervention, definition of lost to follow-up, power statement. When completing the checklist, please use section and paragraph numbers, rather than page numbers.

4) Please include the study protocol document and analysis plan, with any amendments, as Supporting Information to be published with the manuscript if accepted.

5) Please present the safety data for the study including numbers of specific events and whether or not adverse events are thought to be related to the intervention.

DISCUSSION

Please present and organize the Discussion as follows: a short, clear summary of the article's findings; what the study adds to existing research and where and why the results may differ from previous research; strengths and limitations of the study; implications and next steps for research, clinical practice, and/or public policy; one-paragraph conclusion (no subheading).

FIGURES AND TABLES 

1) Please provide titles and legends for all figures and tables (including those in Supporting Information files). 

2) Please define all abbreviations used in each figure/table (including those in Supporting Information files). 

3) Please consider avoiding the use of red and green in order to make your figure more accessible to those with color blindness. 

SUPPLEMENTARY MATERIAL

1) For supplementary figures and tables, please see the general comments under TABLES and FIGURES and amend accordingly.

2) We suggest reporting statistical information as detailed above – see under ABSTRACT. Please be sure to define all numerical values.

3) As for the main manuscript, please indicate whether analyses are adjusted to help facilitate transparent data reporting please also detail the factors adjusted for and present the unadjusted analyses for comparison. If not, please clearly state the reasons why not.

4) Please cite your Supporting Information as outlined here: https://journals.plos.org/plosmedicine/s/supporting-information

REFERENCES

1) PLOS uses the numbered citation (citation-sequence) method and first six authors, et al.

2) Please ensure that journal name abbreviations match those found in the National Center for Biotechnology Information (NCBI) databases (http://www.ncbi.nlm.nih.gov/nlmcatalog/journals), and are appropriately formatted and capitalised.

3) Where website addresses are cited, please specify the date of access (e.g. [accessed: 16/09/2023]).

4) Please also see https://journals.plos.org/plosmedicine/s/submission-guidelines#loc-references for further details on reference formatting. 

Comments from the reviewers:

Reviewer #1: The authors report the results of a placebo-controlled randomized clinical trial of oral resveratrol in adults with osteoarthritis (OA) of the knee. The study was designed appropriately and the manuscript is clearly written. I have no doubt that the results are valid. My only request is that the authors delete P-values in Table 2 for the secondary outcomes. It is not appropriate to perform hypothesis tests on secondary outcomes if the primary outcome does not achieve significance at the 0.05 level. 

Reviewer #2: This manuscript presents the results of a randomised placebo-controlled trial assessing the effect of oral resveratrol in patients with knee osteoarthritis. My comments relate to the statistical aspects of this manuscript. For the most part, the statistical analysis presented in the manuscript reflects that planned in the SAP, however I have a few questions and comments about the analysis and the presentation of results. 

1. What is meant by "first cause of disability"? Is this the most common cause of disability in this group?

2. The "French paradox" is mentioned: the observation of low rates of death from coronary heart disease despite high intakes of cholesterol and saturated fat. Such a paradox, like the "obesity paradox", is likely to be, at least partially, a result of collider stratification bias. See, for example PMID: 37286200; PMID: 24270963. If this paradox is mentioned, that there is a likely selection bias explanation of this paradox must also be mentioned.

3. When the Hussain et al paper is mentioned in the Introduction, the changes within each randomised group are mentioned. It would be more appropriate to describe the change between groups that Hussain et al found when describing this paper, since change within groups is not what randomised trials are designed to detect.

4. The conditional longitudinal data analysis model applied by the authors has been the subject of criticism in the statistical literature - as pointed out in PMID: 20527014. A key criticism holds for the analysis presented here if restricted maximum likelihood (REML) has not been applied: if REML was applied, please state this. If not, please revise your analysis with REML applied.

5. When differences in mean change from baseline were calculated given the results of the cLDA model, were these derived as linear combinations of the estimated parameters from the cLDA model? Please provide a bit more detail in the analysis section. 

6. In the SAP it was mentioned that GEE-based analyses were applied for binary outcomes, but this is not mentioned in the main paper. Please clarify if the planned GEE approach was used and specify what working correlation matrix was assumed. If the planned GEE approach cannot be applied please explain why.

7. Instead of stating that there was no difference between the resveratrol and placebo groups, it is more appropriate to say that the estimated differences between groups were small with wide confidence intervals. A difference between groups was observed - it was just not clinically meaningful was not estimated with a high degree of precision. 

8. Were patients expected to take all supplied caplets? 

9. Please provide a comparison of the baseline characteristics of participants who provided the primary outcome compared to those who did not. The implications of the drop out of patients on the results should be included in the Discussion. Why was only a complete case analysis pre-specified? 

10. In the flow chart, it seems that patients were assessed for eligibility on certain characteristics after been randomly allocated to groups. Is this correct? What happened with the further participation of these patients? 

Reviewer #3: 

In this study, the authors aim to determine whether resveratrol supplementation as an add-on therapy to usual care, could reduce knee pain at 3 months in individuals with painful knee osteoarthritis.

The study holds clear scientific interest as numerous in vitro and animal studies demonstrate a significant effect of this molecule on inflammation. As the authors note, however, there are no clinical studies that have confirmed these findings. Importantly, the authors have modified the galenic form of the molecule, overcoming the low digestive absorption of trans-resveratrol by using it as a dry powder.

The clinical trial design is appropriate and aligns perfectly with the study objectives. It was a double-blind, randomized, placebo-controlled, phase 3 trial conducted in 3 tertiary care centers.

Despite the study's limitations, such as the lengthy recruitment period, the lack of control over other treatments received by the patients or the inability to recruit the anticipated number of patients, the data from the study are interesting because these compounds need to be seriously studied in clinical setting. Indeed, there is very limited information available on this matter.

Minor point:

"but pain intensity reduction was minimal in the placebo group". Was it a real placebo or a meloxicam group?

Reviewer #4: My thanks for the invite to review this thoughtful, well conducted and reported randomised trial of resveratrol vs matched placebo for knee osteoarthritis.

Generally speaking, I find this paper to be of very high quality, and the thoroughness of the research team is apparent in the good study design, appropriate outcomes, and inclusion of additional documentation that help allow scrutiny and therefore boost transparency of this project - the teams' efforts are certainly appreciated.

I have several comments on the paper, but would not hesitate to recommend this paper be published in PLoS Medicine following authors' response to the following (minor) queries/comments:

- This trial is preregistered, and the statistical analysis plan and protocol are i

---

## [Decision Letter · Decision Letter 2]

25 Jun 2024

Dear Dr. Nguyen,

Thank you very much for re-submitting your manuscript "Oral resveratrol in adults with knee osteoarthritis: a randomized placebo-controlled trial (ARTHROL)" (PMEDICINE-D-24-00858R2) for review by PLOS Medicine.

Thank you for your detailed response to the editors' and reviewers' comments. I have discussed the paper with my colleagues and it has also been seen again by two of the original reviewers. The changes made to the paper were mostly satisfactory to the reviewers. As such, we intend to accept the paper for publication, pending your attention to the remaining reviewer and editorial comments below in a further revision. When submitting your revised paper, please once again include a detailed point-by-point response to the editorial comments.

[LINK]

We ask that you submit your revision within 1 week (Jul 02 2024). However, if this deadline is not feasible, please contact me by email, and we can discuss a suitable alternative.

Please do not hesitate to contact me directly with any questions (atosun@plos.org). If you reply directly to this message, please be sure to 'Reply All' so your message comes directly to my inbox.

We look forward to receiving the revised manuscript.

Sincerely,

Alexandra Tosun, PhD

Associate Editor 

PLOS Medicine

plosmedicine.org

Requests from Editors:

GENERAL COMMENTS

1) The Editors agree with Reviewer #4 on the reinstatement of the p-values for the secondary outcomes. Please revise accordingly.

2) Please cite the reference numbers in square brackets. Citations should be preceding punctuation.

COMPETING INTEREST

Please add this statement to the manuscript's Competing Interests: "CN is an Academic Editor on PLOS Medicine's editorial board."

DATA AVAILABILITY STATEMENT 

The Data Availability Statement (DAS) requires revision. Please note that a study author cannot be the contact person for the data. 

ABSTRACT

1) l.59: Please define ‘SD’ at first use.

2) l.61: Please define ‘CI’ at first use.

3) In the last sentence of the Abstract Methods and Findings section, please describe the main limitation(s) of the study's methodology.

4) Please change the sub-heading ‘Interpretation’ to ‘Conclusions’.

5) Abstract Conclusions: Please change to "In this study, we observed that compared with placebo, oral resveratrol did not reduce knee pain in people with painful knee osteoarthritis". You may add a short sentence about the implications for future clinical research.

AUTHOR SUMMARY

1) ll.69-87: Please remove the section ‘Research in context’ and provide the Author Summary.

We ask that you include a short, non-technical Author Summary of your research to make findings accessible to a wide audience that includes both scientists and non-scientists. The authors summary should consist of 2-3 succinct bullet points under each of the following headings:

• Why Was This Study Done? Authors should reflect on what was known about the topic before the research was published and why the research was needed.

• What Did the Researchers Do and Find? Authors should briefly describe the study design that was used and the study’s major findings. Do include the headline numbers from the study, such as the sample size and key findings. 

• What Do These Findings Mean? Authors should reflect on the new knowledge generated by the research and the implications for practice, research, policy, or public health. Authors should also consider how the interpretation of the study’s findings may be affected by the study limitations. In the final bullet point of ‘What Do These Findings Mean?’, please describe the main limitations of the study in non-technical language.

The Author Summary should immediately follow the Abstract in your revised manuscript. This text is subject to editorial change and should be distinct from the scientific abstract. Please see our author guidelines for more information: https://journals.plos.org/plosmedicine/s/revising-your-manuscript#loc-author-summary”

INTRODUCTION

l.88: Please provide a heading for the Introduction section.

METHODS AND RESULTS

1) l.154: We suggest introducing the abbrevation ‘NRS’ here (i.e., “…11-point pain numeric rating scale (NRS)..”).

2) ll.213-217: Please include the two links provided here as references and remove the links from the main text.

3) l.232: Please define ‘WHO’ at first use.

4) l.237: Please define ‘SD’ at first use.

5) ll.257-258: Please write ‘cLDA’ and ‘ANCOVA’ in full (“As a sensitivity analysis, we also analyzed primary outcome with cLDA (constrained longitudinal data analysis) model when considering only baseline and 3 months data and with classical ANCOVA (analysis of covariance).”).

6) l.272: Please write ‘GEE’ in full.

7) ll.281-283: Please remove the funding statement from the main text. The information should only be included in the metadata of the online submission form.

DISUCSSION

1) l.353: Please define ‘HFD’.

2) ll.354-355: Please write 'IL-10' and 'HO-1' and 'PAI-1' in full or add the definition in parentheses after the abbreviation.

3) l.398, please change to: “…COVID-19…”

REFERENCES

PLOS uses the numbered citation (citation-sequence) method and first six authors, et al. Please revise accordingly.

FIGURES

Figure 2: Please consider avoiding the use of red and green in order to make your figure more accessible to those with color blindness. Also, please write ‘NRS’ in full.

SUPPLEMENTARY MATERIAL

1) Thank you for providing the CONSORT checklist. Please replace the page numbers with paragraph numbers per section (e.g. "Methods, paragraph 1"), since the page numbers of the final published paper may be different from the page numbers in the current manuscript.

2) Appendix 5: Please define ‘ANCOVA’

Comments from Reviewers:

Reviewer #2: I thank the authors for their responses to my comments on the previous version of this manuscript. I have only two follow-up comments. 

1. I don't understand what is meant by "a failure to marginalize variance of differences in proportions" when describing why the pre-specified GEE-based approach was not used. Was the issue that random effects for only 3 centers were included? Please provide a clarification of this. 

2. Page 19, line 414: I suggest changing "have low effect on the estimate…" to "would have a minimal impact on the estimate…". Further, I suggest changing "Moreover sensitivity analysis on impact of missing data confirmed that there is no evidence of difference between arms for primary outcome." To "Moreover, our worst case and best case sensitivity analyses indicate that our conclusions are robust to a wide range of assumptions regarding this missing data. "

Reviewer #4: My thanks for the invitation to respond to the revised version of the paper. I see that the authors have addressed the comments from all reviewers adequately and thoughtfully. I wouldn't hesitate to recommend this paper for publication in PLoS Medicine. 

The only minor comment I have for this revised version would be as follows:

Reviewer #1 states "My only request is that the authors delete P-values in Table 2 for the secondary outcomes. It is not appropriate to perform hypothesis tests on secondary outcomes if the primary outcome does not achieve significance at the 0.05 level." - I'd strongly disagree with this assertion. Please could the authors *re-include* the p-values from table 2.

Hypothesis tests of secondary outcomes should not be contingent on the primary outcome. When the primary outcome shows weak evidence of statistical significance, the related secondary outcomes can often provide insight as to why this may have occurred; and to exclude hypothesis tests of secondary outcomes which are unrelated or weakly related to the primary outcome, yet predicated on the primary outcome's statistical significance, is specious. 

To give an example: If the primary outcome of a trial is mortality, it would be very odd to exclude hypothesis tests of the secondary safety outcomes (for example number of serious adverse events, hospitalisations, etc.) based solely on the fact that the mortality outcome was nonsignificant! 

Yours sincerely,

Matthew Parkes

Research Fellow

Centre for Biostatistics

University of Manchester

[LINK]

General Editorial Requests

---

## [Editor Report · Decision Letter 3]

8 Jul 2024

Dear Dr Nguyen, 

On behalf of my colleagues and the Guest Academic Editor, Matthew James Parkes, I am pleased to inform you that we have agreed to publish your manuscript "Oral resveratrol in adults with knee osteoarthritis: a randomized placebo-controlled trial (ARTHROL)" (PMEDICINE-D-24-00858R3) in PLOS Medicine.

I appreciate your thorough responses to the reviewers' and editors' comments throughout the editorial process. We look forward to publishing your manuscript, and editorially there are only a few remaining minor stylistic/presentation points that should be addressed prior to publication. We will carefully check whether the changes have been made. If you have any questions or concerns regarding these final requests, please feel free to contact me at atosun@plos.org.

Please see below the minor points that we request you respond to:

1) Abstract: ll.65-66, we suggest changing to: Our study has limitations in that it was underpowered and the effect size, estimated to be 0.55, was optimistically estimated.

2) Please check whether the Author Summary has been exchanged with the text provided in the file ‘PMEDICINE-D-24-00858R3_Author Summary.docx’.

3) References: For references [40] and [41] (where website addresses are cited), please specify the date of access using the word “accessed”, e.g. “[accessed: 02/07/2024]”.

4) Figure 2: Please write ‘NRS’ in full. We also suggest adding details about the scale, e.g., "The scale is an 11-point numeric rating scale in 10-point increments (0, no pain, to 100, maximal pain)."

5) Table 3: For the 1st, 2nd, and 5th rows, please add statistical information about the number in parentheses, e.g., "Patients with at least one serious adverse event or one minor adverse event (%)," or change the information below the table to "Numbers are absolute frequencies or n (%).”.

6) Appendix 5: Please define ‘NRS’, and ‘CI’ in the list of abbreviations.

PRESS

Sincerely, 

Alexandra Tosun, PhD 

Associate Editor 

PLOS Medicine